# What Is the Impact of Intraoperative Microscope-Integrated OCT in Ophthalmic Surgery? Relevant Applications and Outcomes. A Systematic Review

**DOI:** 10.3390/jcm9061682

**Published:** 2020-06-02

**Authors:** Chiara Posarelli, Francesco Sartini, Giamberto Casini, Andrea Passani, Mario Damiano Toro, Giovanna Vella, Michele Figus

**Affiliations:** 1Ophthalmology, Department of Surgical, Medical, Molecular Pathology and of the Critical Area, University of Pisa, 56126 Pisa, Italy; chiaraposarelli@gmail.com (C.P.); giamberto.c@alice.it (G.C.); andreapassani@gmail.com (A.P.); giovanna.vella28@gmail.com (G.V.); figusmichele@gmail.com (M.F.); 2Department of General Ophthalmology, Medical University of Lublin, 20079 Lublin, Poland; toro.mario@email.it; 3Faculty of Medical Sciences, Collegium Medicum Cardinal Stefan Wyszyński University, 01815 Warsaw, Poland

**Keywords:** intraoperative OCT, microscope-integrated OCT, macular surgery, corneal surgery, cataract surgery, retinal detachment surgery, proliferative diabetic retinopathy surgery, glaucoma surgery, strabismus surgery, paediatric examination

## Abstract

Background: Optical coherence tomography (OCT) has recently been introduced in the operating theatre. The aim of this review is to present the actual role of microscope-integrated optical coherence tomography (MI-OCT) in ophthalmology. Method: A total of 314 studies were identified, following a literature search adhering to the preferred reporting items for systematic reviews and meta-analyses (PRISMA) guidelines. After full-text evaluation, 81 studies discussing MI-OCT applications in ophthalmology were included. Results: At present, three microscope-integrated optical coherence tomography systems are commercially available. MI-OCT can help anterior and posterior segment surgeons in the decision-making process, providing direct visualization of anatomic planes before and after surgical manoeuvres, assisting in complex cases, and detecting or confirming intraoperative complications. Applications range from corneal transplant to macular surgery, including cataract surgery, glaucoma surgery, paediatric examination, proliferative diabetic retinopathy surgery, and retinal detachment surgery. Conclusion: The use of MI-OCT in ophthalmic surgery is becoming increasingly prevalent and has been applied in almost all procedures. However, there are still limitations to be overcome and the technology involved remains difficult to access and use.

## 1. Introduction

Optical coherence tomography (OCT) is a non-contact imaging technique which uses low-coherence interferometry to detect photons backscattered from biological tissues, visualizing a cross-sectional map with micrometre spatial resolution [1].

The first intraoperative OCT use was reported in 2005 by Geerling et al., in which a time domain anterior segment OCT (4 optics, Lübeck, Germany) was coupled with a beam splitter on the front lens of a surgical microscope [2]. However—especially thanks to the advent of Fourier domain techniques—its use in the operating theatre has increased, providing new insight into the surgical management of ocular diseases [3].

The introduction of OCT in the operating theatre began with the development of handheld OCT (HH-OCT) [4], which was first used for ocular imaging in an intensive neonatal care unit by Chavala et al. [5]. It was later introduced in ocular surgery by Dayani et al. in 2009 [6]. HH-OCT represented the first attempt to overcome the traditional chin-rest position for imaging acquisition, allowing imaging in supine patients and sterile settings. However, HH-OCT images had low reproducibility and were affected by motion artefacts. To address these issues, HH-OCT were installed on operating microscopes (microscope-mounted OCT), and the first commercially available systems were the Bioptigen EnVisu (Bioptigen, Research Triangle Park, NC/Leica, Wetzlar, Germany) and the Optovue iVue (Optovue, Fremont, CA, USA) [3]. However, these devices required placement over the patient’s eye, such that the microscope had to be displaced and surgery paused [7]. This procedure could have altered the sterility of the surgical field [8]. Therefore, there was still a need for an efficient, operator-independent, usable, integrated solution [7].

In 1998, Boppart described the first microscope-integrated OCT (MI-OCT) [9]. However, the OCT and the microscope did not share any optics, reducing the working distance and potentially obstructing the surgical workflow. Moreover, the acquired images were displayed on a monitor, which was not visible to the surgeon during the operation [10]. Later MI-OCT designs integrated the OCT into the microscope optics, providing the opportunity to visualize a two-dimensional OCT section of the surgical field into the microscope’s oculars (named the heads-up display) or on an external display. These new devices ensured imaging reproducibility without the necessity for new space and could be permanently integrated into the microscope [11]. 

Between 2014 and 2015, three commercial MI-OCT systems were cleared by the Food and Drug Administration and became available: the Haag-Streit iOCT-system (Haag-Streit, Koeniz, Switzerland), the Zeiss platform Rescan 700 (Zeiss, Oberkochen, Germany), and the Leica EnFocus system (Leica, Wetzlar, Germany) (Figure 1) [3].Two scanning modes were available: filming and snapshots. Any type of anaesthesia was suitable for MI-OCT scanning, although complete akinesia was advisable. Important prerequisites to obtain high-quality imaging of the posterior segment were clear optic media and normal intraocular pressure [8].

These instruments have been proven to be feasible in the operating theatre, as well as safe and useful in decision-making processes [12,13].

This systematic review aims to analyse the data available on the impact of MI-OCT in ophthalmic surgery and patient outcomes; current limitations and recent advancements in MI-OCT technology are also summarized.

## 2. Experimental Section

A comprehensive search of the PubMed database was performed on 22 April 2020. A combination of subject headings and text words were used, as needed, to define MI-OCT. Keywords used for the search were “intraoperative OCT eye”, “intraoperative-OCT”, “intraoperative optical coherence tomography eye”, “iOCT”, “microscope-integrated OCT”, “microscope optical coherence tomography”, and “MI-OCT”. The search workflow was designed in adherence to the preferred reporting items for systematic reviews and meta-analyses (PRISMA) statement (Figure 2) [14].

The process applied for this review consisted of a systematic search of all available articles regarding the impact and the advantages related to the use of MI-OCT in ophthalmologic surgery and patient outcomes, both for the anterior and posterior segments of the eye.

All identified electronic data captured were independently evaluated, in terms of their titles and abstracts, by two reviewers (C.P. and F.S.) to identify relevant articles. In addition, the references of identified articles were manually checked to find any potential studies relevant for review purposes. All studies available in the literature reporting original data on MI-OCT were initially included without restriction for study design, sample size, and intervention performed. Review articles, articles written in languages other than English, and ex vivo studies were excluded from the present review.

The same reviewers selected the studies according to the inclusion and exclusion criteria. Any disagreement was assessed by consensus and a third reviewer (F.M) was consulted when necessary. The following data, using an Excel spreadsheet, were evaluated by two reviewers (C.P. and F.S.) independently: study title, author, year of publication, study design, number of participants, ocular assessments, and outcomes. For unpublished data, no effort was made to contact the corresponding authors. All the selected records were evaluated to define the strength of evidence, according to the Oxford Centre for Evidence-Based Medicine (OCEM) 2011 guidelines and the Scottish Intercollegiate Guideline Network (SIGN) assessment system for individual studies, as implemented in the Preferred Practice Patterns of the American Academy of Ophthalmology [15,16]. Finally, the quality of evidence was also assessed, based on the Grading of Recommendations Assessment, Development, and Evaluation (GRADE) system [17].

## 3. Results

A total of 314 studies were identified following the initial literature search. After elimination of duplicated records, 288 abstracts were identified for screening and 113 of these met the inclusion and exclusion criteria for full-text review. Thirty-one articles were excluded: 19 were not written in English, 9 were review papers, and 3 were ex vivo studies. After full-text evaluation, 82 studies discussing the use of MI-OCT in ophthalmology were included [13,18,19,20,21,22,23,24,25,26,27,28,29,30,31,32,33,34,35,36,37,38,39,40,41,42,43,44,45,46,47,48,49,50,51,52,53,54,55,56,57,58,59,60,61,62,63,64,65,66,67,68,69,70,71,72,73,74,75,76,77,78,79,80,81,82,83,84,85,86,87,88,89,90,91,92,93,94,95,96,97,98].

The included studies were grouped considering the application of MI-OCT during anterior or posterior segment surgery, as listed in the Appendix A. In particular, 40 studies evaluated the application of MI-OCT during anterior segment surgery, 38 evaluated the application of MI-OCT in posterior segment surgery, and 4 evaluated the application of MI-OCT in both segments.

Furthermore, 38 were prospective studies, 22 were retrospective studies, and 20 were case reports. We also included a post-hoc analysis and a phase 2 clinical trial.

The following outcomes were assessed regarding MI-OCT assisted surgery: the feasibility and main advantages in glaucoma, cornea, cataract, and strabismus surgeries for anterior segment procedures and feasibility and main advantages in macular surgery, retinal detachment surgery, and proliferative diabetic retinopathy treatment for posterior segment surgery. Finally, the feasibility and main advantages in paediatric examination were evaluated.

The current systematic review reports a qualitative analysis, detailed issue-by-issue in narrative fashion for the heterogeneity of available data and the design of the available studies (i.e., case reports or case series). 

### 3.1. MI-OCT and Anterior Segment Surgery

MI-OCT has been used in various anterior segment procedures, such as glaucoma filtering procedures, deep anterior lamellar keratoplasty (DALK), Descemet membrane endothelial keratoplasty (DMEK), Descemet stripping automated endothelial keratoplasty (DSAEK), penetrating keratoplasty (PK), cataract surgery, and strabismus surgery.

#### 3.1.1. Glaucoma Surgery

MI-OCT found its first application in glaucoma surgery, in which it can assist the surgeon in filtering and non-filtering procedures. In trabeculectomy, MI-OCT allows for visualization of the exact postsurgical anatomy of the sclera, Schlemm’s canal, trabecular meshwork, and anterior chamber [18]. In particular, it can verify scleral flap depth and thickness and determine the completeness and extent of the trabecular stromal ostium. These are critical steps in achieving a functional bleb, which is not easily checkable (especially for novel surgeons) [19]. In Ahmed glaucoma valve implantation, MI-OCT assists the surgeon in tube intrascleral passage and tube positioning in the anterior chamber, visualizing the actual distance from the cornea and the iris [19]. In bleb revision and needling, thanks to a detailed assessment, MI-OCT can detect the location and extent of adhesion, simplify their breakage both above and below the scleral flap, and visualize large single hyporeflective cavity development at the end of the procedure [20]. In addition, MI-OCT plays a role in confirming trabecular meshwork removal in ab-interno trabeculotomy with microhook or Trabectome (NeoMedix Corporation, Tustin, CA, USA) and can localize Schlemm’s canal during canaloplasty procedures. Its main limits in ab interno glaucoma surgery are represented by the MI-OCT wavelength and the absence of oblique scanning options [21,22,23,24]. Finally, MI-OCT could avoid the use of an intraoperative gonioscopy lens for goniosynechiolysis, visualizing the peripheral anterior synechiae extension, synechiolysis adequacy, and angle opening extension at the end of the procedure [19].

#### 3.1.2. Cornea Surgery

MI-OCT has been largely applied in corneal transplant surgery, in particular in the DISCOVER study (Determination of Intraoperative Spectral Domain Microscope Combined/Integrated OCT Visualization During En Face Retinal and Ophthalmic Surgery)—a single-site, multi-surgeon, prospective study that evaluated the role of MI-OCT in ophthalmic surgery [25,26]. In this study, three integrated OCT systems were used: Zeiss RESCAN 700, Bioptigen/Leica EnFocus, and a prototype developed by the Cole Eye institute. A total of 837 eyes were enrolled, 244 for the anterior segment arm and 593 for the posterior one. OCT imaging was obtained in 97.8% of cases; unsuccessful imaging acquisition was due to hardware/software malfunction, poor view precluding image acquisition, and surgeon’s decision. In particular, 123 DSAEK procedures were performed (with 50% anterior segment cases), 59 DMEK (24% of cases), and 10 DALK (10% of cases). MI-OCT altered the surgeon’s decision-making in 43.4% of anterior segment cases. Its feedback was fundamental in evaluating graft-host apposition, assessing the presence of fluid at the interface in DSAEK and DMEK, and verifying graft orientation in DMEK, allowing surgeons to abandon the use of an “S” stamp on corneal tissue. In 43.4% of cases (106 patients), MI-OCT modified surgical decision-making; in particular, considering graft position or persistence of fluid. In anterior segment procedures, real-time feedback was the favoured imaging modality and the preferred review strategy was external screen display, compared to microscope oculars [13].

Considering DALK, successful big-bubble formation depends on the depth reached by the cannula tip from the internal corneal surface. In the DISCOVER study, MI-OCT confirmed the big-bubble formation already noted clinically in three of ten cases; in two of ten cases, it guided additional manoeuvres as the bubble was incomplete, and in three of ten cases, it helped the surgeon in manual stromal dissection by confirming the dissection depth [13]. Specifically, the average depth in successful cases is lower than that in failed ones, which have an increased risk of conversion to PK (90.4 μm versus 136.7 μm, respectively), requiring a manual deep lamellar dissection [99]. Therefore, MI-OCT aids the surgeon to assess the needle depth within the corneal stroma, guiding re-trephination in the case of very low or very deep depths. It also confirms the big-bubble dissection plane and complete Descemet’s membrane (DM) bareness after trephination, as well as evaluating the residual stromal depth in the cornea bed [13].

MI-OCT can also help in challenging circumstances and in the management of adverse events. Singh et al. described a case of a 22-year-old man with gelatinous drop-like corneal dystrophy, where MI-OCT guided manual stromal dissection in DALK without microkeratome assistance [27]. Furthermore, Selvan et al. observed the development of a triple chamber following DALK procedure: one chamber between the donor lenticule and the host’s Dua layer, one between the latter and the host’s DM, and the true anterior chamber [28]. Residual interface viscoelastic was recognized as the cause and MI-OCT was used to guide its drainage. Intracameral air tamponade ensured the vanishing of the two extra chambers [28]. Finally, Sharma et al. reported a case of DM detachment following DALK, attributed to microperforations in the DM that occurred during manual dissection [29]. Thanks to continuous MI-OCT monitoring, the authors injected 20% SF6 intracameral, under the DM, and performed additional incisions through the anterior corneal surface. In the end, MI-OCT was used to show complete DM attachment with fluid disappearance at the graft-DM interface [29].

Regarding DSAEK, MI-OCT can visualize all surgical steps, such as graft insertion, unfolding, and attachment, supporting the surgeon especially in the case of oedematous cornea [30]. It also allows quick donor tissue visualization, avoiding the mixing of different kind of donor tissues when multiple transplant procedures are scheduled in a single theatre in a day [100]. MI-OCT does not slow the surgical workflow, increases imaging stability, and improves surgical decision-making by identifying residual fluid interfaces (undetectable by operative microscope alone), which may be related to an increased rate of graft dislocation or detachment [31]. In the DISCOVER study, persistent interface fluid was visualized on MI-OCT in 46 (54.8%) of 84 patients, in which the surgeon clinically presumed graft well-attachment [13]. Graft apposition during DSAEK may be significantly improved with controlled pressure elevation and corneal sweep [101]. Titiyal et al. conducted a prospective study on 30 eyes to evaluate the timing of lenticule apposition in sutureless DSAEK performed in clear corneas [32]. In the first group (*N* = 10), in order to facilitate lenticule apposition, a continuous positive intracameral pressure was maintained for 8 min, after which a centrifugal external corneal massage was performed until resolution of the interface fluid. The anterior chamber was decompressed after 13 min of intracameral positive pressure. In the second group (*N* = 10), the external corneal massage was performed immediately after intracameral positive pressure, lasting for 8 min. In the third group (*N* = 10), external corneal massage and intracameral positive pressure were performed (as in group 2) but just for 5 min; then, the anterior chamber was decompressed. MI-OCT showed the complete resolution of fluid interface within 3 min in both groups 2 and 3. Thus, simultaneous positive intracameral pressure and external corneal massage minimize the time for graft apposition, preventing prolonged intraocular pressure elevation and reducing surgical time [32]. 

Moreover, the same authors proposed a novel sign to confirm correct graft orientation in ultra-thin DSAEK. When the graft is well-attached with the stromal side up, the lenticule bevelled edge forms an acute angle with the overlying stroma, named the “acute-angled bevel sign”. This is especially useful in hazy cornea, where it is hard to confirm graft orientation [33].

DMEK, compared to DSAEK, allows a faster visual rehabilitation and a better visual outcome; however, it is more challenging [102]. In particular, when the donor lenticule is very thin, there are several obstacles to overcome, such as graft preparation, graft visualization, unfolding in the host’s anterior chamber, and risk of postoperative graft detachment [103]. Nevertheless, MI-OCT, by providing real-time cross-sectional images, helps surgeons to visualize and assess the graft orientation. It also enables fast graft positioning (6.1 ± 3.0 min), reducing graft manipulation (especially in oedematous cornea) [34]. Therefore, after graft insertion, MI-OCT has been used to confirm lenticule apposition and reduce the intracameral positive pressure period, without compromising clinical outcomes [35]. 

Sharma et al. performed 25 DMEK procedures in severe oedematous cornea [36]. MI-OCT allowed for placement of the main incision away from posterior stromal irregularities due to previous surgery, reducing the rate of wound leakage due to poor wound apposition. MI-OCT was also used to identify areas of incomplete descemetorhexis and avoid unnecessary DM scraping. Moreover, it can be used to precisely localize any area of graft fold, allowing the surgeon to decide whether to treat them or not. Finally, MI-OCT can confirm graft orientation and complete apposition (Figure 3) [36].

A report of the outcomes of the first 100 DMEK procedures in the DISCOVER study described a reduced unfolding time and rebubbling rate using MI-OCT, compared to the average values reported in the literature about DMEK, which was not assisted by MI-OCT, 4.4 min versus 5.7–6.4 min and 6.4% versus 28.8%, respectively [37]. Likewise, Saad et al. reported a rebubbling rate of 7% in a case series of 14 DMEK assisted by MI-OCT [34]. Finally, MI-OCT could have training purposes, helping surgeons that have already performed DSAEK to learn the DMEK procedure [13,38].

In PK, MI-OCT can indicate irido-corneal adhesion and help surgeons to avoid iris trauma during initial trephination [39]. It can also guide peripheral synechiolysis, detecting non-clinically evident ones and confirming their complete resolution [40].

In laser-assisted corneal procedures such as laser phototherapeutic keratectomy (PTK), MI-OCT can assess corneal transparency and superficial roughness; however, further studies are needed to clarify whether this could translate to benefit patients postoperatively [41].

Moreover, MI-OCT allows us to measure in vivo riboflavin penetration in accelerated collagen cross-linking in patients affected by keratoconus. In particular, the mean riboflavin penetration was 149.39 ± 15.63 μm in the epi-on technique and 191.04 ± 32.18 μm in the epi-off technique [42]. Additionally, MI-OCT can help surgeons in Bowman layer transplantation, a novel technique for keratoconus patients who have a cornea that is too steep or too thin and cannot undergo ultraviolet corneal crosslinking or intracorneal ring segments [104].

Interestingly, two other applications of MI-OCT are corneal trauma and keratoconus acute corneal hydrops. In the first case, MI-OCT can be used as an adjunctive tool for intraoperative assessment, enabling the identification of the corneal dissection plane; meanwhile, it can also verify any cataract injuries, especially capsular defects [43]. In case of acute corneal hydrops, the current treatment is to wait for spontaneous resolution of the corneal oedema, and after, a keratoplasty is performed (if necessary). Nevertheless, MI-OCT can accelerate resolution. In particular, in the case of small DM defects, in the acute phase, MI-OCT can guide the drainage of intrastromal fluid pockets, followed by anterior chamber with SF6 tamponade and pre-descemetic sutures [44,45]. In the case of larger defects, the surgeon can perform mini-DMEK (3–5 mm graft diameter), monitoring the graft unfolding and adhesion with MI-OCT [46].

MI-OCT can also play a role in keratoprosthesis implantation, allowing correct assembly and positioning of the device, which are the cornerstones in obtaining satisfying visual results. It can also highlight any gap between the backplate and graft or anterior optic and graft, which are usually not visible intraoperatively with a microscope alone. Therefore, MI-OCT can reduce the learning curve, potential risks, and complications [47].

Finally, other minor applications for MI-OCT are corneal biopsy, corneal ulcer debridement with tuck-in multi-layered amniotic membrane transplantation, corneal Intacts (Addition Technology, Lombard, IL, USA) placement, lamellar keratotomy for Salzmann degeneration, and small incision lenticule extraction, in the case of difficult lenticule extraction [13,48,49,50]. In the latter case, MI-OCT can provide real-time visualization of lenticule relation with anterior stromal cap and stromal bed [50].

#### 3.1.3. Cataract Surgery

In cataract surgery, MI-OCT has found its main application in training novel surgeons. It can be used to visualize corneal incisions, confirm lens position, and assess trenching depth during phacoemulsification, preventing any iatrogenic capsular rupture. At the end of the surgical procedure, it indicates the adequacy of stroma hydration, decreasing the rate of postoperative wound leak and hypotony (Figure 4) [18,105]. These benefits apply both to micro-incision cataract surgery (MICS) and femtosecond laser-assisted cataract surgery (FLACS) [51]. In particular, Titiyal et al. published a study about corneal incision in 129 patients undergoing phacoemulsification; 77 with MICS and 52 with FLACS [52]. The first group showed a significant increase in incision-site DM detachment postoperatively. This detachment solved spontaneously without consequence for visual outcome one month after surgery. Reviewing MI-OCT images, an irregular proximal corneal incision was detected more frequently in the MICS group (87.1% versus 16.3%, *p* < 0.001), which was identified as the most important predictive factor for incision-site DM detachment [52].

MI-OCT can also be helpful for expert surgeons in complicated cases, such as identifying a capsular defect in traumatic cataract or a true posterior polar cataract, when clinically suspected [51]. 

Anisimova et al., using MI-OCT, were able to identify the Berger’s space between the posterior lens capsule and the anterior hyaloid in 21 of 28 eyes [53]. They observed, within the space, the presence of hyper-reflective dots and particles of different shapes and sizes corresponding to lens micro-fragments, cellular material, or medical suspension. The authors noticed also an increased risk of posterior lens capsule rupture by surgical tips in the case of Wieger ligament discontinuity. In this case, this space became excessively hydrated during phacoemulsification, leading to anterior displacement of the posterior capsule [53].

Titiyal et al. evaluated 50 senile white cataracts with MI-OCT, proposing a new classification system and management strategy for each type of cataract [54]:Type I cataract, with regularly arranged cortical fibres without raised intra-lenticular pressure (ILP).Type II cataract, with anterior cortical convexity and raised pre-operative ILP, but without fluid release and ILP resolution at the begin of capsulorhexis.Type III cataract, with intralenticular clefts combined with areas of homogeneous ground glass appearance, raised pre-operative ILP with fluid release, and partial ILP resolution at the begin of capsulorhexis.Type IV cataract, with homogeneous ground glass appearance of the anterior lens cortex, raised pre-operative ILP, fluid release, and complete ILP resolution at the begin of capsulorhexis.

In a type II cataract, there is an increased risk of rhexis extension, and the authors suggested performing bimanual aspiration/irrigation until ILP lowering is observed with MI-OCT. However, in types III and IV, there is a mild–moderate risk of capsulorhexis extension and so the above-mentioned procedure is not required [54].

Furthermore, MI-OCT can also assess the intraocular lens (IOL) position at the end of phacoemulsification, as shown by Lytvynchuk et al. [55]. In particular, they noticed that contact between the IOL central optic and the posterior capsule rarely occurs. Nevertheless, this contact is supposed to slow the migration of lens epithelial cells, preventing posterior capsule opacification. Therefore, they suggested an improvement of the IOL design, in order to grant immediate contact between the IOL central optic and posterior capsule [55].

Regarding implantable collamer lens (ICL), Titiyal et al. measured ICL vaulting intraoperatively in 40 eyes, highlighting a significant correlation between intraoperative and postoperative values (paired sample correlation 0.954; *p* < 0.001) [56]. MI-OCT displays real-time surgical manoeuvres, preventing any accidental lens touching and facilitating the detection of excessive ICL vaulting intraoperatively [56].

Finally, MI-OCT can be helpful in patients affected by ectopia lentis, identifying any adherence between the anterior lens capsule and corneal endothelium (especially when the cornea is oedematous), preventing it from further damage when the lens is surgically removed [57].

#### 3.1.4. Strabismus Surgery

MI-OCT can also play a role in strabismus surgery, in particular, in identifying rectus muscle insertion—which is usually hard to locate in patients who have had previous surgery—and confirming scleral suture passage [58]. Pihlbald et al. compared extraocular muscle (EOM) measurements obtained with different OCT devices with calliper distance measured during strabismus surgery in 19 paediatric patients [59]. In particular, the OCT devices used were Bioptigen (Leica Microsystems Inc., Buffalo Grove, IL, USA), Spectralis HRA + OCT with Anterior Segment Module (Heidelberg Engineering, Heidelberg, Germany), Visante (Carl Zeiss, Oberkochen, Germany), and Zeiss RESCAN 700. All of them could accurately localize EOM insertion; however, MI-OCT (Zeiss RESCAN 700) succeeded in many cases and provided real-time information during surgery [59].

### 3.2. MI-OCT and Posterior Segment Surgery

Thanks to its impressive results in understanding changes in retinal anatomy during surgery without causing significant time-wasting, the role of MI-OCT has been investigated in almost every field of posterior segment surgery [9,18,60,61]. It offers high intra- and inter-observer reproducibility, especially considering lamellar macular holes, vitreomacular traction (VMT), and epiretinal membranes (ERM) [62]. It is also compatible with common chromovitrectomy dyes and tamponades [18]. MI-OCT may assess the self-sealing of the sclerotomy at the end of pars plana vitrectomy [8].

#### 3.2.1. Macular Surgery

Considering vitreomacular interface disorders, Falkner-Radler et al.—who developed one of the first MI-OCT prototypes—stated that MI-OCT was comparable with retinal dyes in confirming the success of membrane peeling [63]. In particular, using MI-OCT in a cohort of 70 patients (51 epiretinal membranes, 11 vitreomacular traction, and 8 macular hole), it was possible to perform membrane peeling without using dyes in 40% of cases and choosing tamponade according to intraoperative imaging [63]. 

Similar evidence was obtained by Pfau et al., who reported that MI-OCT provided additional information in 74.1% of 32 cases [18]. In particular, it altered the surgical decision in 41.9% of patients regarding cleavage site identification for ERM or ILM peeling, removal of clinically undetectable vitreous remnants, and tamponade choice [18]. Moreover, Leisser et al. observed that MI-OCT helped to complete ERM peeling without the use of chromovitrectomy dyes in a majority of cases (63%) [64]. However, ILM staining was still necessary, as MI-OCT is not able to visualize the posterior hyaloid and the inner limiting membrane intraoperatively [65]. When compared to standard OCT, MI-OCT image quality is lower, being affected by corneal transparency (which is usually reduced during surgery) and longer optical pathways through the operating microscope. Nevertheless, MI-OCT offers immediate visualization of the retinal anatomy during peeling procedures, helping to understand retinal changes due to surgical manipulation [64].

Regarding retinal changes due to membrane peeling, 34 eyes were analysed in the DISCOVER study, where 21 presented a full-thickness macular hole and 13 an epiretinal membrane. MI-OCT intraoperatively detected expansion of the ellipsoid zone to retinal pigmented epithelium distance (3 eyes), subretinal fluid accumulation (1 eyes), and hyper-reflectivity of the inner retinal layers associated with retinal haemorrhage (10 eyes) [66]. In addition, Leisser et al. observed that, three months after ERM peeling, iatrogenic subfoveal and extrafoveal hyporeflective zones were visible only in 7% of cases in a series of 171 patients [67]. Although the clinical outcomes were similar between patients with or without these findings, only a small minority of them developed new deep microscotomata, as seen in microperimetry [68]. Furthermore, a randomized prospective trial on 69 patients published by the same group identified that the presence of intraoperative subfoveal hyporeflective zones was associated with postoperative intraretinal cystoid changes (44.4%) at three months follow-up, compared to patients without intraoperative subfoveal hyporeflective zones. Nevertheless, this difference was not statistically significant. The only risk factor identified for postoperative intraretinal cystoid changes after peeling of idiopathic epiretinal membranes among patients randomized for balanced salt solution and air-tamponade was the presence of pre-operative intraretinal cystoid changes [69].

The main results about the feasibility and advantages of MI-OCT in macular surgery were reported in the three-year results of the DISCOVER study, including 593 eyes in the posterior segment group. Information provided by MI-OCT altered surgical decision in 29.2% of cases. Membrane peeling was performed in 272 patients for different indications (e.g., epiretinal membrane, macular hole, proliferative vitreoretinopathy). In particular, after peeling, imaging revealed residual membranes in 19.8% of patients; on the other hand, MI-OCT prevented unnecessary additional surgical manoeuvres in 40% of cases where the surgeons thought there were epiretinal residual membranes. Moreover, after posterior hyaloid elevation, MI-OCT identified the presence of a possible full-thickness macular hole or outer retinal hole development in 15.8% of cases. Interestingly, over the years, the preference of surgeons for posterior segment procedures has shifted towards a stop and shot approach, rather than real-time feedback. Screen display, compared to microscope oculars, has been confirmed as the preferred review strategy [13,61].

The evidence of the DISCOVER study on macular hole formation during surgery for vitreomacular interface syndrome has been confirmed in other reports, modifying surgical approach and tamponade choice [70,71]. 

In full-thickness macular hole surgery, it has been postulated that macular hole closure begins with inner retinal contraction, forming a roof over a hyporeflective space in the subfoveal space, mimicking a foveal retinal detachment. No pre-operative factors, such as macular hole stage, hole width, or symptom duration, have been correlated with the persistence of hyporeflective subfoveal area [106,107]. Thanks to MI-OCT, Kumar and Yadav proposed a novel intraoperative sign predictive for idiopathic macular hole closure, named the “hole-door sign”: residual tissue fragments at the edges of the hole projecting into the vitreous cavity after ILM peeling. All eyes with this sign have a 100% rate of type 1 closure (presence of outer nuclear layer at the closed macular hole), compared to 60% of eyes without it [72]. However, Inoue et al. reported that the presence of intraoperative residual fragments (the hole-door sign) in type 1 macular hole closure was correlated to limited postoperative visual acuity, compared to type 1 closure without residual fragments [73]. The presence of these fragments has been significantly related to the size of the hyper-reflective inner retinal lesions and associated with pre-operative ERM presence (*p* = 0.04) or with epiretinal proliferation (*p* = 0.007). Apparently, the wound-healing process of the inner retina in closed macular holes seems in contrast with poor visual recovery; a possible explanation may be the damage of the outer retina caused by the contractile properties of the residual fragments [73]. 

MI-OCT can also confirm correct ILM inverted flap positioning, even after complete air–fluid exchange, avoiding the needs of OCT imaging in the immediate postoperative period; thus, it may represent a useful tool for reducing the time of face-down position after surgery (Figure 5) [74,75]. 

Finally, in the case of refractory macular hole, full-thickness neurosensory retinal autograft can be used. In particular, MI-OCT helps in graft sizing and positioning, which are essential for restoring macular structure and improving visual acuity [76,108].

#### 3.2.2. Retinal Detachment Surgery

A post-hoc analysis of the DISCOVER study evaluated MI-OCT-assisted retinal detachment repair. A total of 103 eyes were included—51 uncomplicated primary cases and 52 complicated ones, due to presence of pre-operative proliferative vitreoretinopathy, giant retinal tear, panuveitis, or recurrent retinal detachment. MI-OCT altered the surgical decision in 12% of cases; however, it offered valuable feedback in 50% of complex cases and in 22% of uncomplicated ones. Re-operation rates were 6% in uncomplicated cases and 25% in complex ones at 12 months follow-up; an excellent result compared to the rates reported in the literature, ranging from 13% to 30% [77].

In rhegmatogenous retinal detachment, MI-OCT documented that perfluoron does not affect the integrity of the ellipsoid zone in macula-off retinal detachment [78]. Moreover, it revealed residual submacular fluid after air–fluid exchange despite the use of perfluoron, direct drainage, or drainage retinotomy. This endorses the strict face-down positioning for at least 24 h after surgery, even if submacular fluid was not clinically evident [79,80]. However, there was no correlation between residual submacular fluid detected by MI-OCT and functional or anatomical outcomes, as reported by Obedi et al. [80]. Finally, the real-time feedback provided by MI-OCT may help surgeons to evaluate and treat proliferative vitreoretinopathy or to identify and remove chronic subretinal perfluoron (Figure 6) [13,81].

MI-OCT may also play a role in non-rhegmatogenous retinal detachment, as observed by Lytvynchuk et al. in a patient with morning glory syndrome [82]. In this case, retinal reattachment was obtained only with air endotamponade, revealing strong vitreous adhesion and traction above the macula and the optic disc [82].

Finally, if retinotomy is performed and results are ineffective due to retinoschisis, MI-OCT can highlight residual outer retinal layer adherence to the retinal pigment epithelium, such that the surgeon can extend the incision to release the retinal traction [83].

#### 3.2.3. Proliferative Diabetic Retinopathy Surgery

In the case of proliferative diabetic retinopathy, the DISCOVER study reported that MI-OCT offered real-time and valuable feedback to the surgeon in 50.6% of cases and altered the surgical plan in 26% of cases, especially in detecting retinal holes/breaks, highlighting the need for more membrane peeling or the use of a specific tamponade. Additionally, in the case of vitreous haemorrhage (which precludes pre-operative OCT macula evaluation), MI-OCT detected ERM, vitreomacular traction, or macular oedema, helping the surgeon to decide whether to treat them simultaneously or later. Moreover, MI-OCT has been used to confirm complete ERM removal in the case of vitreoschisis and differentiate retinal detachment and retinoschisis [84].

MI-OCT allows surgeons to perform cystotomy for recurrent diabetic cystoid macular oedema, an experimental treatment proposed by Asahina et al. [85]. At six months of follow-up, central retinal thickness, macular volume, and best-corrected visual acuity improved significantly in 13 eyes (65%) out of 20 affected by recurrent diabetic cystoid macular oedema treated with ERM and ILM peeling and cystotomy [85].

In case of severe proliferative diabetic retinopathy, MI-OCT aids the surgeon in identifying a potential space to safely introduce the vitrector between the surface of the retina and the fibrovascular proliferation. Once a full-thickness retinal hole is achieved, a viscodissection or hydrodissection can be performed to expand the space between the retina and fibrovascular membrane. Afterwards, the proliferative vitreoretinopathy can be safely cut and removed [86].

#### 3.2.4. Other MI-OCT Applications

MI-OCT represents a potentially useful tool for highly myopic eyes, especially for the evaluation of posterior vitreous adhesions after surgically induced vitreous detachment, identification of residual fragments during ERM or ILM peeling, and the presence of macular holes [87]. In high-myopic patients affected by retinoschisis, MI-OCT endorses fovea-sparing ILM peeling, reporting a lack of microarchitectural foveal alterations and significant best-corrected visual acuity improvement, compared to a complete ILM peeling without fovea sparing [88,89]. In the case of myopic traction maculopathy and macular hole-associated retinal detachment, MI-OCT can identify the vitreoschisis and the vitreomacular traction or confirm the closure of the hole by the ILM flap [90].

In uveitis-related vitreoretinal surgery, MI-OCT can assess the fluocinolone acetonide implant placement and provide valuable feedback in subretinal and choroidal biopsies. In addition, in uveitis-related retinal detachment, it altered the surgical decision in 48% of patients, largely related to additional membrane peeling [91].

MI-OCT can also be useful in endophthalmitis, where vitreous opacity prevents pre-operative OCT examination. In particular, it may show diffuse retinal oedema and identify fibrous tissues, allowing their removal without causing retinal tears [92]. 

In addition, other applications of MI-OCT are subretinal injections for gene therapy, positioning of Argus II retinal prostheses, and guiding chorioretinal biopsies. In particular, subfoveal injection of AAV2-REP1 seems to be an increasingly popular treatment for patients affected by genetically confirmed advanced choroideremia, in order to achieve a sustained improvement or maintenance of visual acuity. Thanks to MI-OCT, proper subretinal injection is achievable, avoiding the suprachoroidal plane, excessive foveal stretching, and macular hole formation [93,94].

Regarding Argus II retinal prostheses, MI-OCT can be used to visualize the array/retina interface, which is crucial for implant positioning and functioning. MI-OCT has been shown to be more efficient, compared to HH-OCT [95].

Moreover, intraoperative MI-OCT can verify the proper depth and position of chorioretinal biopsy and evaluate whether the retinal edges are flat after the procedure [96].

### 3.3. MI-OCT and Paediatric Examination

Paediatric examination is another important field of MI-OCT application, especially in the case of new-borns and infants, whose examination is typically challenging due to reduced co-operation, but which is necessary and fundamental to prevent amblyopia. It is usually performed under general anaesthesia and is based on surgical microscopy, allowing funduscopic and gonioscopic evaluation but only if optical media are clear. To date, imaging is limited to ultrasound biomicroscopy, which has a lower tissue resolution compared to OCT and requires an experienced ophthalmologist for reliability. MI-OCT, instead, allows the visualization of all relevant anterior segment structures, even in opaque corneas, and provides corneal thickness and optic disc nerve fibre layer thickness. Furthermore, MI-OCT can detect congenital central superficial hypertrophic scars or corneal keloid, mimicking Peter’s anomaly [97]. MI-OCT can also assess cornea and macula status after trauma. In particular, Coppola et al. reported a case of a paediatric patient with a massive hyphema following domestic trauma [98]. Once the anterior chamber was cleaned, MI-OCT was used to evaluate cornea and macula, in order to exclude immediately pathological findings. This prevented the need for surgical revision, if any damage was diagnosed postoperatively [98].

## 4. Discussion

The role of MI-OCT has rapidly evolved over time. The first published studies were mainly aimed at demonstrating its feasibility in examination and surgeon’s impressions.

Regarding anterior segment surgery MI-OCT-relevant applications are lamellar corneal transplantation (DALK, DSAEK, and DMEK), enabling better visualization of the operating field and surgical instrument. Thanks to this, the surgeon is able to overcome difficulties related to the surgical technique, reduce the time of tissue manipulation, and achieve correct graft orientation and better adhesion. In addition, real-time feedback facilitates the decision-making process and improves surgical technique for both novel and expert surgeons. MI-OCT is also useful in complicated corneal cases and trauma. All these advantages lead to a better clinical outcome with shorter recovery time and reduced re-operation rate.

Based on the encouraging results in corneal surgery, the use of MI-OCT has also spread to cataract surgery, where novel surgeon training and the management of complicated cataracts are the main fields of application. Furthermore, a new classification system for cataracts based on MI-OCT images has been proposed, in order to avoid intraoperative complications during capsulorhexis. Finally, MI-OCT has showed that contact between the IOL central optic and posterior capsule occurs rarely, highlighting the need for IOL design improvement. 

Furthermore, glaucoma surgery could benefit from intraoperative microscope-integrated tomography, especially in the case of opaque optic media. Ab externo and ab interno procedures may be guided by MI-OCT, where bleb morphology and management can be improved through better visualization of the anatomical planes.

Posterior segment surgery could be regarded as a fundamental area which can benefit from intraoperative real-time anatomical structure visualization. Epiretinal membrane visualization and removal could be implemented by MI-OCT, in order to avoid unnecessary surgical manoeuvres and manage accidental macular hole formation. In particular, these devices can detect the safest place to begin ERM peeling and reduce the use of potentially toxic retinal dyes. The understanding of retinal changes subsequent to surgical manipulation may positively impact patient’s visual outcomes and their rates of re-intervention. Regarding macular holes, alteration of hole geometry can determine alterations in the outer retina which may be relevant in the clinical outcome. Intraoperative imaging can confirm ILM inverted flap positioning, preventing the need for postoperative OCT.

Both rhegmatogenous and non-rhegmatogenous retinal detachments benefit from MI-OCT, with an inferior re-operation rate, favourably impacting the clinical outcome. Furthermore, complex case management is facilitated.

The same considerations are valid for patients affected by proliferative diabetic retinopathy, where surgical plane identification and macular visualization are crucial for success, especially in the case of vitreous haemorrhage. 

Highly myopic eyes, uveitic patients, endophthalmitis, placement of Argus II implant, subretinal biopsy, and subretinal gene therapy are other promising fields of MI-OCT application.

Finally, MI-OCT represents an essential tool for paediatric examination, especially in new-borns and infants, where imaging is traditionally limited to ultrasound biomicroscopy, an operator-dependent technique with low resolution.

This systematic review includes 82 manuscripts on MI-OCT; however, the published reports are mainly prospective (38) and retrospective (22) studies of low or very low level and quality of evidence. Case reports (20) were also included because, in our opinion, they may add relevant information which cannot be overlooked. Another issue is represented by the number of patients included in studies reported; usually less than 30 patients. There are also confounders and biases related to the wide variety of conditions and procedures reported, such that it was not possible to perform any data synthesis.

To date, no randomized or controlled trials have been performed, and the typical follow-up period is limited: The longest report of evidence is the three-year results of the DISCOVER study. 

In addition, surgeon’s awareness of using MI-OCT may have influenced feedback reports. The presence of a trained assistant in the operating theatre during image acquisition and surgeon feedback collection may represent another bias. These issues should be considered in data interpretation.

The largest study published on MI-OCT is the DISCOVER study, with more than 800 eyes involved. It demonstrated the feasibility and the usefulness of MI-OCT in the surgical decision-making process, showing that anterior and posterior segment surgeons have different preferences for OCT acquisition and visualization strategies. The former preferred real-time images, as working “under” the tissue of interest is less influenced by shadowing by a surgical instrument, while the latter preferred static imaging, mainly because the metallic instruments involved prevent retinal surface visualization. Viewing images on an external display is the preferred review modality both for anterior and posterior segment surgeons. This preference has increased during the follow-up period, as it allows for the detection of subtle changes and details that are not visible in images displayed in microscope oculars.

The DISCOVER study results confirmed similar findings from a preceding large Prospective study on Intraoperative and perioperative Ophthalmic imagiNg with optical coherEncE tomography: the PIONEER study. In the PIONEER study, the researchers evaluated the feasibility, safety, and utility of a microscope-mounted OCT—the Bioptigen SDOIS portable spectral domain OCT (SD-OCT) system (Bioptigen, Research Triangle Park, NC)—installed on Leica and Zeiss microscopes. A total of 531 eyes were enrolled (275 anterior segment cases and 256 posterior segment cases) and OCT images were obtained in 98% of cases, requiring a median time of 4.9 min per case, with no adverse effects. Thanks to a better understanding of the underlying tissue configuration, intraoperative OCT modified surgeon’s impressions and impacted on surgical care in 48% and 9% of lamellar keratoplasty cases and in 43% and 8% of membrane peeling procedures, respectively [12]. Nevertheless, microscope-mounted OCT enhanced stability but did not allow for real-time intraoperative OCT. In particular, this mode of image acquisition requires stopping the surgery to displace the surgical microscope and position the OCT device over the patient’s eye.

Further studies with randomized controlled trials are needed to better characterize patient outcomes, surgical efficiency, and safety, although several indicators already attest that this technology may improve patient care.

### Future Perspectives

Huge progress must be carried out to introduce OCT imaging into the operating theatre; nevertheless, in order to become a must-have device, other steps need to be done, particularly regarding tracking systems, OCT-compatible instruments, and real-time feedback.

Specifically, three-dimensional (3D) surgical tip tracking systems and automated real-time quantitative metrics have been developed for anterior segments, significantly improving the surgeon’s ability to reach a target depth and facilitating cornea procedures. An analogous system for anterior chamber angle and posterior segment evaluation could be developed in the near future [109,110].

Another rising field is OCT-compatible instruments, in order to avoid excessive light shadowing phenomena during OCT acquisition, precluding real-time surgical manipulation. In particular, among six semi-transparent materials, Ehlers et al. selected polycarbonate to realize a surgical pick, retinal forceps, and a corneal needle, allowing the visualization of several surgical manoeuvres such as membrane peeling, corneal penetration, and vessel manipulation in model eyes [111]. The concept was to create instrument using a semi-transparent material from the tip to allow OCT signal transmission [112].

Moreover, Horstmann et al. developed a custom-built high-resolution MI-OCT system that can detect corneal lymphatic vessels [113]. These are precocious clues for corneal neovascularization, which is the principal limiting factor for corneal transplant survival. Lymphatic vessels are usually not clinically visible; therefore, this device opens up the possibility of non-invasive monitoring of corneal lymphangiogenesis, allowing the implementation of proactive therapy to extend graft survival [113].

Contrast-enhanced MI-OCT is another nascent field, improving the visualization of tissue interfaces. The use of different agents in porcine eyes, such as triamcinolone, prednisolone, lipid-based artificial tear, and indocyanine green, has been described. Future developments could aid understanding of intraocular fluidics or wound stability [114,115].

Recently, Ehlers et al. combined MI-OCT imaging with a 3D surgical visualization system (NGENUITY™, Alcon. Fort Worth, TX, USA), evaluating the feasibility and surgeon’s feedback of this new visualization platform [116]. Seven patients were included (3 ERM, 2 macular hole, 1 symptomatic vitreous opacity, and 1 proliferative vitreoretinopathy). Two-dimensional OCT images were obtained in all cases and shown on a 4K resolution screen. Surgeons and assistants reported improved visualization both of the OCT and of the surgical field. However, further studies are required, in order to compare the conventional microscope-based approach with the digital 3D screen [116].

Recently, swept-source OCT (SS-OCT) has been added to MI-OCT at Duke University in 2016 by Toth et al. [117,118,119]. SS-OCT and SD-OCT are both Fourier domain detection techniques; the former uses a tuneable swept laser (with wavelength of 1050 nm) with a single photodiode as a detector, while the former uses a near-infrared superluminescent diode (wavelength of approximately 840 nm) with a spectrometer as a detector [120]. These technological improvements allow faster, deeper, and larger scans, including choroid layers, as well as overcoming the minimum acquisition lag which still persists in commercial MI-OCT technologies. This technology has been named four-dimensional (volumes over time) MI-OCT (4D-MI-OCT) and offers immediate and intuitive OCT feedback [121]. It is capable of recording live 3D images, which can be visualized directly into the surgeon’s oculars, providing a truly three-dimensional visualization of volumes in real-time using a novel dual-channel stereoscopic heads-up display. Some recent applications of 4D-MI-OCT are visualization of suture depth in strabismus surgery, evaluation of big-bubble formation in DALK, subretinal delivery of stem cells, and assessment of fibrovascular membranes and residual tractions in vitrectomy for proliferative diabetic retinopathy, especially in the cases of tractional retinal detachment, combined tractional-rhegmatogenous retinal detachment, and vitreous haemorrhage [122,123,124,125,126,127]. Moreover, 4D-MI-OCT has been used to perform transvitreal vitrectomy-assisted retinochoroidal biopsy in the case of uveal melanoma. In particular, it has been used to confirm adequate penetration depth of the cutter and the absence of subretinal fluid surrounding the retinotomy after cutter removal [121]. Finally, another recent 4D-MI-OCT application is Argus II positioning, specifically to assess array–tissue apposition, which is crucial for optimal functioning of the device itself [128].

Thanks to swept-source technology, another experimental MI-OCT system was able to perform intraoperative optical coherence tomography angiography in two children with retinal vascular diseases. It allowed for detailed analysis of vascular pattern comparted to fluorescein angiography, visualizing small pathological retinal vessels usually obscured by fluorescein staining due to pre-existing laser scars [129].

MI-OCT provides incomparable real-time cross-sectional imaging at very high-resolution, but the burden of manual device manipulation and intraoperative interpretation can be challenging for surgeons, especially while performing surgical manoeuvres. To address this issue, device manipulation by voice commands has been developed to change the MI-OCT scan location [130]. Brooks et al. merged ancillary imaging technology into a single-screen viewing platform to improve intraoperative interpretation (named digitally assisted vitreoretinal surgery, DAVS) [131]. In particular, it is a high-definition visualization system displaying the surgical field in real-time, as well as imaging from MI-OCT, ocular endoscopy, and electronic health records/imaging, onto a 3D High-Definition flat-panel screen (NGENUITY™, Alcon. Fort Worth, TX, USA). This prototype is more user-friendly, more ergonomic, and conveys more precise information, compared to image visualization on different displays [131]. In particular, surgeons reported a significant lower rate of back discomfort and headaches, compared to the conventional MI-OCT visualization system: 1.0% vs. 18.7% (*p* < 0.0001) for back pain and 5.2% vs. 20.9% (*p* < 0.002) for headaches, respectively [132].

Furthermore, MI-OCT has been integrated into an intraocular robotic interventional surgical system, opening the possibility for semi-automated lens extraction. The robotic OCT guided system was able to perform automated cataract extraction in 30 cadaveric pig eyes [133].

Finally, MI-OCT could be spread to other surgical specialities, such as general surgery and neurosurgery, thanks to the possibility of recording micron-scale resolution imaging without tissue surface contact [134,135]. Furthermore, a special magnetomotive MI-OCT has been developed, which modulates the magnetic field within the tissue during OCT imaging. In this way, it can detect magnetic nanoparticles (magnetite or maghemite) in biological tissue at microscale, finding possible clinical applications in tumour extent definition and treatment [136].

## 5. Conclusions

At present, based on the relevant published literature, the principal application of MI-OCT seems to be helping surgeons in the decision-making process, avoiding unnecessary manipulations. In this way, anterior and posterior segment surgeries could be safer (e.g., reducing iatrogenic lesion) and faster (e.g., avoiding time-consuming procedures). Beside the surgeon’s advantages, MI-OCT has a significant impact even on patient outcomes, reducing re-operation rates or individualizing the best postoperative positioning of the patient. Thanks to the wider availability of these devices, it is easier to perform paediatric examinations in new-borns and infants with opaque medias. MI-OCT could also have didactic and training purposes, helping novel surgeons in their approach to cataract surgery. Some issues still exist which limit the widespread adoption and utilization of these devices, mainly their high cost, followed by the technical improvements needed, such as a three-dimensional tracking systems and contemporary visualization of OCT imaging and the surgical field. Ongoing and future research showing further benefits in the care and prognosis of patients will be extremely important to justify the cost and to facilitate the diffusion of MI-OCT in daily clinical practice.

## Figures and Tables

**Figure 1 jcm-09-01682-f001:**
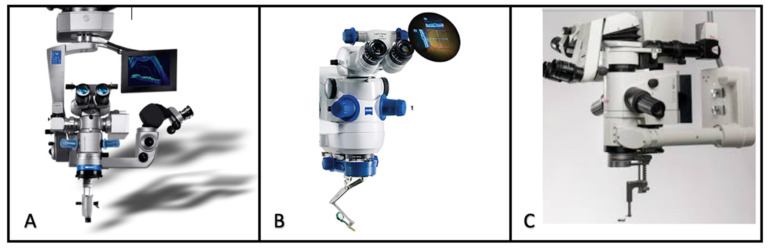
Commercial MI-OCT systems. (**A**) Haag—Streit Surgical iOCT system, (**B**) Zeiss RESCAN 700 system, and (**C**) Leica EnFocus system.

**Figure 2 jcm-09-01682-f002:**
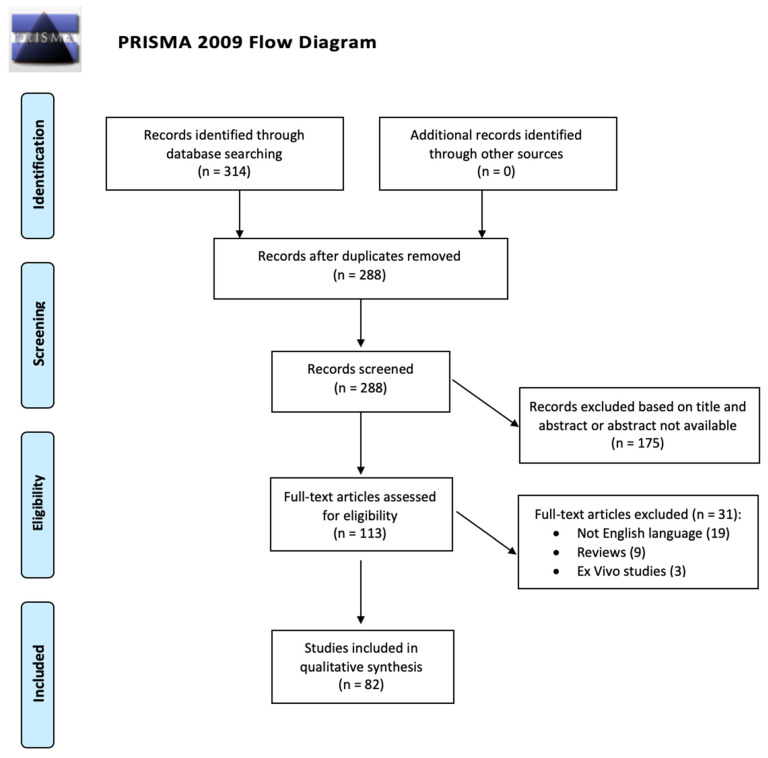
Preferred reporting items for systematic reviews and meta-analyses (PRISMA) flowchart.

**Figure 3 jcm-09-01682-f003:**
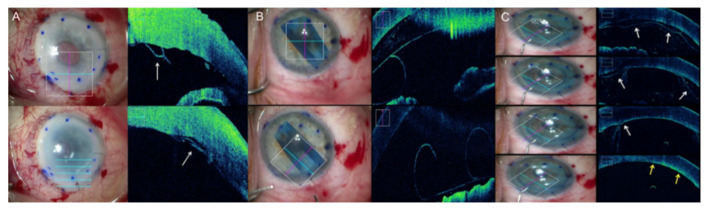
Microscope-integrated optical coherence tomography (MI-OCT) assisted Descemet’s membrane endothelial keratoplasty (DMEK). (**A**) The left column shows en-face view. The right column shows the accompanying MI-OCT image. Top row: Peripheral Descemet’s membrane remnants prior to graft insertion (arrow). Bottom row: Attached graft overlapping Descemet’s membrane remnants (arrow). (**B**) Top row: Inverted graft after insertion. Bottom row: Corrected graft orientation. (**C**) From the top to bottom, rows represent consecutive time points. Progressive graft apposition during gas infusion (fluid interface indicated by white arrows; complete graft apposition indicated by yellow arrows).

**Figure 4 jcm-09-01682-f004:**
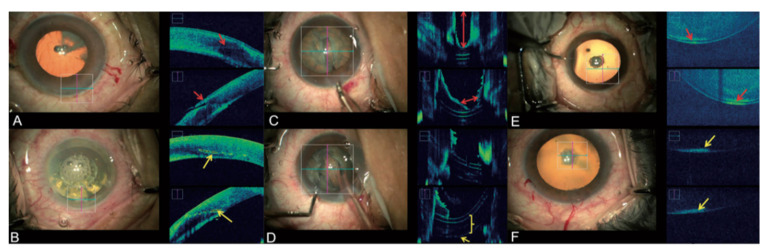
MI-OCT assisted Cataract surgery: The left column shows en-face view. The right column shows the accompanying MI-OCT image. (**A**) Manual keratome corneal incision (red arrows). (**B**) femtosecond laser-assisted corneal incision (yellow arrows). (**C**) Depth and width of trench before dividing the nucleus (red arrows). (**D**) Image after nucleus division (posterior capsule, yellow arrow), epinuclear plate (yellow bracket). (**E**) Pseudo-posterior polar cataract: a gap between lens and posterior capsule (red arrows). (**F**) True posterior polar cataract: no gap between lens and posterior capsule (yellow arrows). Reproduced with the permission from reference [58].

**Figure 5 jcm-09-01682-f005:**
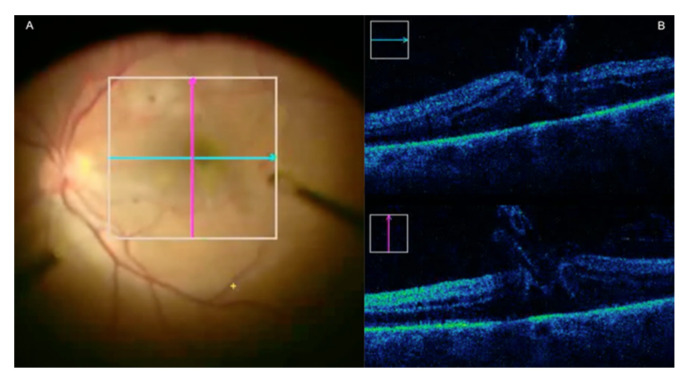
MI-OCT assisted Macular Hole surgery. (**A**) En-face view. (**B**) MI-OCT images confirm inverted ILM flap positioning after fluid-air exchange. Reproduced with the permission from reference [83].

**Figure 6 jcm-09-01682-f006:**
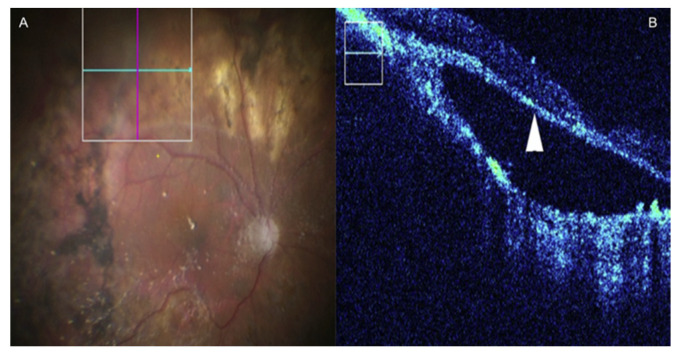
MI-OCT assisted Retinal detachment surgery. (**A**) En-face view. (**B**) MI-OCT shows a subretinal membrane consistent with proliferative vitreoretinopathy (arrowhead). Reproduced with the permission from reference [87].

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
