# Peer review of "What Is the Impact of Intraoperative Microscope-Integrated OCT in Ophthalmic Surgery? Relevant Applications and Outcomes. A Systematic Review"

_jcm, 2020, doi:10.3390/jcm9061682_

Round 1

Reviewer 1 Report

Thank you for choosing me to be a reviewer

My comments and thought while reading this paper:

Intro

  1. The “A first prototype was developed in 2005 by Geerling et al., which coupled an OCT with a beam 40 splitter on the front lens of a surgical microscop” should come prior to “Fourier….
  2. The introduction is interesting but does not mention the aim of this study -why to do this review? What is the purpose? How will this help? What is the question the review will answer? What new information?

Methods

  1. In addition to the PRISMA protocol, you could add the Scottish Intercollegiate Guideline Network (SIGN) assessment system for individual studies as implemented for Preferred Practice Patterns by the American Academy of Ophthalmology. I used it here https://www.ncbi.nlm.nih.gov/pubmed/27743157. Which is better this or the oxford?

Results

  1. Table 1 should be a supplemental
  2. You should summarize the data for ease of reading. Maybe try “bubble chart” where the size of the point is the number of N. I used if before here https://www.ima.org.il/FilesUploadPublic/IMAJ/0/381/190922.pdf
  3. Comprehensive review- are there comparative studies? The use of OCT versus not using it? This could be in a separate table

Conclusion

  1. Please add the price of these machines and if you think the cost worth it.
  2. Which machine is better?

Reviewer 2 Report

First review dedicated to development and application of the intraoperative OCT was published as a book chapter:
Lytvynchuk L, Glittenberg C, Binder S. (2017) Intraoperative Spectral Domain Optical Coherence Tomography: Technology, Applications, and Future Perspectives. In: Meyer CH, Saxena S, Sadda SR (ed.) Spectral Domain Optical Coherence Tomography in Macular Diseases. Springer, New Delhi, India, pp 423-443.

This book chapter shall be mentioned in the references and discussed.

Reviewer 3 Report

Drs. Posarelli et al. have adequately described the history, the need for, and the utility of MI-OCT. Their methods are sound, and they provide very useful results for the ophthalmic surgeon, helping him (or her) to make decisions about the potential utility of MI-OCT for their practice or surgical setting. I also do not detect flagrant plagiarism nor do there appear to be any conflicts of interest. 

I do have major concerns about this manuscript. The topic is very interesting, and I can understand what the authors are communicating, but there are too many instances of non-standard English. I could only get through pg. 14 (of 26 pages [not including references] to review).  I have included suggestions for those first two(2) pages of edits (almost all are grammar edits), but please have this paper English-editing service before I review again. # - minor editorial;* - grammar/English writing

  1. Pg. 1, line 13: #(minor) - add "(OCT)" after "...tomography"
  2. Pg. 1, line 25: * This is an awkward sentence. Please re-format; I suggest: "However, there are limits to this technology's utility and availability."
  3. Pg. 1, lines 27-29: * remove capital letters at the beginning of each keyword
  4. Pg. 2, end of line 34: * add "a"; change to "visualizing a cross-sectional"
  5. Pg. 2, line 36: * please remove new paragraph here
  6. Pg. 2, line 39: * please remove new paragraph here
  7. Pg. 2, line 41: #(minor) - consider changing the word "started" to "began"
  8. Pg. 2, lines 42-43: * - consider reformatting sentence to "It was first used for ocular imaging by …"
  9. Pg. 2, line 43: #(minor) - consider changing "Afterwards," to "It was later introduced in ocular surgery by..."
  10. Pg. 2, line 44: * - consider changing "allowing it in supine patient and sterile settings..." to "allowing its use in supine patients and sterile..."
  11. Pg. 2, line 46: * - add "s" to end of "artefact"
  12. Pg. 2, end of line 47: #(minor) - consider continuing the sentence as ", and the first commercially available systems were..."
  13. Pg. 2, near end of line 49: #(minor) - consider changing "Nevertheless, these devices..." to "However,these devices..."
  14. Pg. 2, lines 51-52: * - Consider changing to "Therefore, there was still a need for an efficient, operator-independent, usable integrated solution."
  15. Pg. 2, line 53: #(minor) - Consider new sentence here, after "(MI-OCT)". "However, the OCT and microscope..."

As you can see, there are too many English/grammar edits to continue the review.  Please have this manuscript extensively edited for English, and I would love to review again.

Reviewer 4 Report

The authors summarize an extensive amount of data about the microscope-integrated OCT. It would be very helpful for readers who want to overview the literature about it. 

Since it involves a vast amount of data, it would be nicer if it could be more organized in a simple way. For example, it would be better to overview Table 1 if the data were divided into subsections, such as 'glaucoma', 'cornea', and 'cataract'.  

Otherwise the review is helpful to overview and compare existing publications.

Round 2

Reviewer 3 Report

My main objections to the manuscript were the English edits required. The authors have remedied these.

Therefore, my previous ratings (excellent/outstanding) for the significance, organization, and science stand, and I have upgraded my ratings for English style and readability to above average/excellent.  I have some final suggestions.

Line 72: "analyse" is not standard English. Please change to "analyze"

Line 337: please add "=" or ":" between "correlation" and "0.954"